# Fast and flexible processing of large FRET image stacks using the FRET-IBRA toolkit

**Gautam Munglani** * , **Hannes Vogler** , **Ueli Grossniklaus**

Department of Plant and Microbial Biology and Zürich-Basel Plant Science Center, University of Zürich, Zürich, Switzerland

* gmunglani@gmail.com

## Abstract

Ratiometric time-lapse FRET analysis requires a robust and accurate processing pipeline to eliminate bias in intensity measurements on fluorescent images before further quantitative analysis can be conducted. This level of robustness can only be achieved by supplementing automated tools with built-in flexibility for manual ad-hoc adjustments. FRET-IBRA is a modular and fully parallelized configuration file-based tool written in Python. It simplifies the FRET processing pipeline to achieve accurate, registered, and unified ratio image stacks. The flexibility of this tool to handle discontinuous image frame sequences with tailored configuration parameters further streamlines the processing of outliers and time-varying effects in the original microscopy images. FRET-IBRA offers cluster-based channel background subtraction, photobleaching correction, and ratio image construction in an all-in-one solution without the need for multiple applications, image format conversions, and/or plug-ins. The package accepts a variety of input formats and outputs TIFF image stacks along with performance measures to detect both the quality and failure of the background subtraction algorithm on a per frame basis. Furthermore, FRET-IBRA outputs images with superior signal-to-noise ratio and accuracy in comparison to existing background subtraction solutions, whilst maintaining a fast runtime. We have used the FRET-IBRA package extensively to quantify the spatial distribution of calcium ions during pollen tube growth under mechanical constraints. Benchmarks against existing tools clearly demonstrate the need for FRET-IBRA in extracting reliable insights from FRET microscopy images of dynamic physiological processes at high spatial and temporal resolution. The source code for Linux and Mac operating systems is released under the BSD license and, along with installation instructions, test images, example configuration files, and a step-by-step tutorial, is freely available at github.com/gmunglani/fret-ibra.

## Author summary

FRET is a fundamental imaging technique used to generate fluorescence signals sensitive to molecular conformations and interactions. Despite its wide use and the large body of literature on the theoretical steps required to process images generated from this procedure, we were unable to locate a tool that contained the entire processing workflow, whilst

**Data Availability Statement:** The source code for Linux and Mac operating systems is released under the BSD license and, along with installation instructions, test images, example configuration

files, and a step-by-step tutorial, is freely available at github.com/gmunglani/fret-ibra.

**Funding:** This work was supported by the University of Zurich (UG), the Research and Technology Development Project MecanX of SystemsX.ch, the Swiss Initiative in Systems Biology (to UG), and, in part, by Swiss National Science Foundation grant CR22I2_166110 (to UG). The funders had no role in study design, data collection and analysis, decision to publish, or preparation of the manuscript.

**Competing interests:** The authors have declared that no competing interests exist.

allowing the user the flexibility to adjust parameters for maximum accuracy and runtime efficiency. FRET-IBRA was thus created to be an all-in-one, open-source, parallel solution to process FRET images, while eliminating complications arising from repeated image format conversions. Besides enhancing the background subtraction algorithm for FRET images, several additional options were implemented for the users to improve the quality of the signal for their specific use case. FRET-IBRA is primarily built for flexibility when handling large image stacks by supporting sequences of image frames to be treated independently, greatly reducing time spent on splitting and concatenating image stacks. In accuracy and speed benchmarks against more general background subtraction packages, FRET-IBRA was able to provide the cleanest results with a fast runtime, leading to reliable analysis without additional tuning.

This is a *PLOS Computational Biology* Software paper.

## Introduction

Ratiometric biosensors based on FRET (Förster Resonance Energy Transfer) are often used to quantify the dynamics of physiological processes at subcellular resolution. Several methodologies have been developed to optimize the calibration of these protocols [1, 2], but the amount of data generated still requires fast, scalable, and flexible software tools for efficient and robust analyses.

Accurate retrospective background subtraction and photobleaching correction algorithms are critical for the extraction of precise spatio-temporal pixel intensity distributions from time-lapse images. This is particularly true for studies relying on the time series analysis of ion concentrations linked with cell kinematics [3]. The inherent variability of a single experiment over longer time scales necessitates the use of heuristically defined process parameters that can be applied to specific data splits rather than the entire image stack to combat temporal drift [4].

Several algorithms have been implemented as stand-alone packages or within existing tools (*ImageJ*, *CellProfiler*, and *qTfy*) to facilitate the efficient processing of a variety of microscopic imaging modalities. Traditional methods are typically very robust and widely applicable but often use a static process parameter set, which is not trivial to customize for specialized applications [5–7]. A number of high-performance algorithms designed for retrospective background subtraction in light microscopy images, using clustering and constrained minimization methods, have been developed to improve precision. However, they have been shown to be generally unsuitable for characterizing growing cells by time-lapse images, which exhibit high temporal correlation [4, 8–10].

In this work, we describe FRET-IBRA (Image Background-subtracted Ratiometric Analysis), a fully parallelized, configuration file-based tool developed to simplify the ratiometric analysis of FRET image stacks [11]. FRET-IBRA accepts multi-image TIFF stacks with different bit-depths (8, 12, and 16 bit) as input, and outputs both multi-image TIFF and HDF5 stacks for further downstream analyses. To provide maximum flexibility, this tool incorporates the processing of discontinuous frames within an image stack as well as the correction of individual frames with optimal parameters to ensure that temporal noise is minimized. It should be noted that the ratiometric approach described here is limited to linked constructs of

fluorescent proteins where pixelwise stoichiometry is maintained. To showcase the performance of this tool, FRET images of growing *Arabidopsis thaliana* pollen tubes are processed to create ratio stacks to evaluate the spatial distribution of calcium ions driving growth.

## Design and implementation

### Description

FRET-IBRA consists of three modules which are responsible for (i) the background subtraction of acceptor and donor time-lapse images, (ii) the optional registration and alignment of these processed image stacks in case a dual-view setup was used to produce ratio images, and (iii) the photobleaching correction, respectively (Fig 1A). Furthermore, the tool outputs several metrics to gauge the performance of the background subtraction algorithm on individual frames, allowing for the fast detection and reprocessing of sub-optimally processed frames. Logging and HDF5 storage capabilities are also provided to aid in reproducibility.

### Modules

The background subtraction module tiles the image frame into squares of a set dimension and uses the DBSCAN clustering algorithm with an Euclidean metric to identify tiles as foreground or background (Fig 1B and 1C). This algorithm is adapted from Schwarzfischer and colleagues (2011) [8], using an expanded feature space, which includes the median and higher moments (standard deviation, skewness, and kurtosis) of the tiles' pixel intensity distribution along with the position of its intensity-weighted centroid (Fig 1C). The background tile intensities are used to quadratically interpolate over the entire grid to estimate the spatially varying background intensity caused by shading effects. The resulting background intensity is then subtracted from the original image (Fig 1B). This method requires that, for the DBSCAN algorithm, a tuning parameter $\epsilon$ is provided in the configuration file. $\epsilon$ is defined as the maximum distance, in feature space, between two tiles to be classified as the same cluster. This algorithm is effective in correcting spatial shading and removing local noise along with providing accurate background subtraction for images with few objects of interest, but is optimized for single-cell images. When dealing with images with very low signal intensities, users should carefully inspect the corrected images before proceeding as very low signals can produce artefacts (brachypodium_roots example in the FRET-IBRA repository). This module outputs an animation of frame-by-frame metrics of algorithm performance for visual inspection as well as the filtered HDF5 dataset and TIFF image stacks.

The ratiometric processing module contains several optional pre-processing steps before the ratio image is produced. Image registration between the donor and acceptor stacks is first performed by rigid linear transformation, followed by the application of a small kernel bilateral smoothing filter. Otsu's thresholding is then used to binarize the images, after which singularities at the image boundary are removed. The percentage of non-zero (foreground) pixels per frame of the donor and acceptor stacks (Fig 1D) and the ratio of median intensity to bit depth (Fig 1E) are provided as a quantitative metric of the algorithm's performance over the entire image stack. Sharp discontinuities in these metrics between sequential frames may indicate that specific frames require reprocessing with a different $\epsilon$. The ratiometric images, which represent the ratio of the acceptor and donor image stacks, are saved as a HDF5 dataset and a TIFF image stack.

Photobleaching correction is then optionally performed on frames of the acceptor and/or donor stacks, using a regularized linear or exponential fit on the median frame intensity (Fig 1E). The frame range used for the fit is specified in the configuration file and applied to all frames starting with the onset of photobleaching, defined as the lower bound of the provided

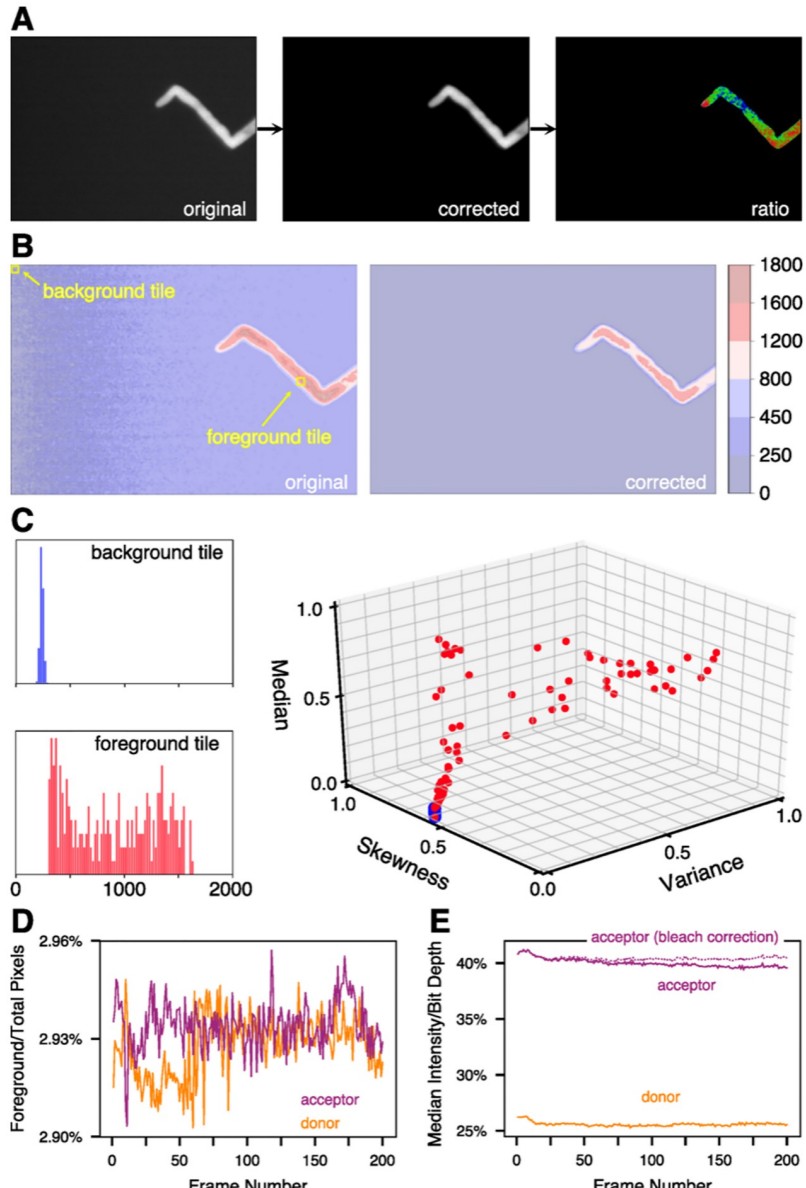

**Fig 1. FRET-IBRA workflow.** A) Image evolution from the original image to the background-corrected image and the final ratiometric image. B) A contour map of the pixel intensities of the original and corrected images. C) Pixel intensity distributions of example background and foreground tiles shown in B), along with the reduced feature space showing median, skewness, and variance, with each point representing a tile. The blue cluster indicates the background tiles, while the red cluster indicates foreground tiles. D) The percentage of foreground pixels can be used to control the quality of the background subtraction for each channel. E) The median signal intensity of donor and acceptor channel allows for analysis and correction of bleaching effects, and serves as a an additional quality indicator for background subtraction.

frame range. The corrected ratiometric images are saved as an output TIFF image stack, while the bleach correction factors are stored in the HDF5 dataset for further analysis, if required.

FRET-IBRA is built to process images with few objects of interest that are densely packed, allowing for accurate background estimation. As the background subtraction algorithm classifies tiles solely as foreground or background, images with a wide spatially homogenous

distribution of cells or other objects of interest might not be processed correctly due to poor estimation of the background signal.

## Results

To showcase the efficacy of FRET-IBRA, FRET images displaying the calcium distribution in growing *Arabidopsis thaliana* pollen tubes were corrected with the toolkit and the result was compared with existing background subtraction tools like BaSiC [10] and the background sub- tractor module of the Mosaic image processing package [12] (Fig 2A). The optimal back- ground window size parameter in FRET-IBRA, which determines how many square tiles the image width should be divided into, is determined to be 40 for this image stack. This parame- ter requires tuning based on the image feature dimensions, to ensure accurate foreground extraction without utilizing excessive computational resources. Furthermore, it should be emphasized that the higher the number of tiles, the smaller the tile size and the longer the run- time of the background subtraction algorithm.

As can be seen by two line samples of the image pixel intensities in Fig 2B, FRET-IBRA is the most effective package for subtracting background pixel intensities whilst maintaining the peak intensity value of the pollen tube. Furthermore, in addition to this background subtrac- tion, FRET-IBRA smoothens the resulting background pixel intensities by reducing the stan- dard deviation of the background noise distribution. With 4 cores, the runtime for FRET-IBRA lies between BaSiC and Mosaic (Fig 2C). However, unlike the other packages, the parallel implementation of FRET-IBRA allows for further runtime reduction that scales line- arly with an increasing number of cores.

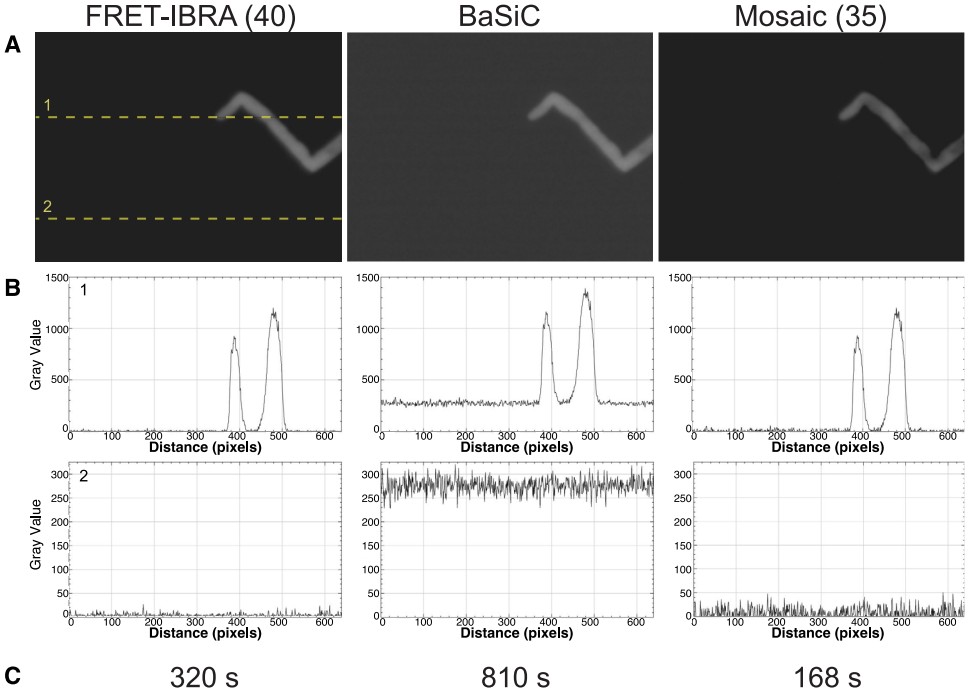

**Fig 2. Comparison of FRET-IBRA with existing background subtraction plugins for ImageJ (BaSiC [10] and the background subtractor module of the Mosaic suite [12]).** A) Frame 1 of the donor (CFP) channel after background subtraction. The dashed lines indicate the rows of pixels analysed in B). B) Pixel intensities after background subtraction. Top and bottom panels correspond to lines 1 and 2, respectively. C) Runtime for the background subtraction for 1000 frames. All tests were run on 4 cores with an Intel Core i7 (Quad-Core, 2.8GHz, 16GB RAM) computer.

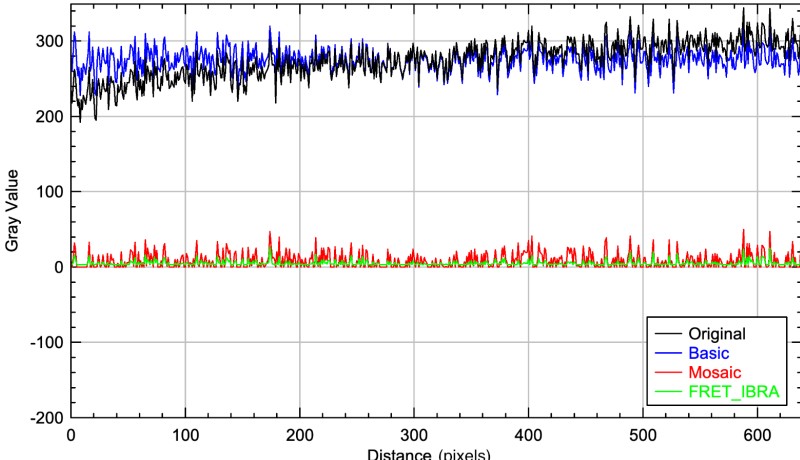

**Fig 3. Background subtraction comparison.** Background levels according to dashed line 2 in Fig 2 after background subtraction with FRET-IBRA, BaSiC, and Mosaic compared with the original image. The BaSiC plugin corrected only the shading, while Mosaic and FRET-IBRA corrected the shading and subtracted the background. Overall, processing the image with FRET-IBRA results in the background with the lowest pixel intensity and standard deviation.

The BaSiC tool is shown to be effective in shading correction of the image when compared to the original; however, the average background pixel intensity remains mostly unaltered (Fig 3). In contrast, the background subtractor module of Mosaic successfully subtracts the intensity of the background pixels, although the standard deviation of the resulting background noise remains relatively high (Table 1). While the window size for Mosaic was set at 35, it should be noted that this parameter does not seem to have a significant effect on the accuracy of this tool.

Ratiometric image stacks were then produced for FRET-IBRA, BaSiC, and Mosaic from the processed acceptor and donor stacks (Fig 4). The resultant ratio images were scaled to display the full 8-bit range, and the first frame processed by each tool was used for comparative purposes (Fig 4A). The background signal in the BaSiC-processed ratio image was uniformly high, which generally allowed for good discrimination between foreground and background signal, with the exception of a lower intensity halo surrounding the pollen tube outline (Fig 4A, top panel). Visually, the dynamics (signal spectrum) of the foreground can be seen to suffer from high background values, rendering any quantitative analysis very difficult. The ratio image produced by Mosaic was significantly better due to the resulting high signal to noise ratio of the input images (Fig 4A, middle panel). The acceptable background intensity, however, was diminished by irregular noise with occasional high intensity peaks that could not be completely eliminated with additional filtering, along with the continued presence of a halo

**Table 1. Pixel intensity statistics after running background subtraction algorithm with variable window sizes.** The window size in FRET-IBRA defines the number of tiles the image width should be divided into, i.e., for a 640x480 pixel image, a window size set at 40 results in 40 windows along the width and 30 windows along the height (1200 windows with a tile size of 16x16 pixels). The window size cannot be set for BaSiC. Parameters used for our comparison are marked yellow.

| Pixel Intensity | FRET-IBRA | | | Mosaic | | | BaSiC |
|---|---|---|---|---|---|---|---|
| Window | 20 | 40 | 80 | 20 | 35 | 70 | - |
| Mean | 5.35 | 4.92 | 5.33 | 8.41 | 8.24 | 8.73 | 275.21 |
| Max | 41 | 27 | 10 | 51 | 50 | 50 | 320 |
| Min | 1 | 2 | 4 | 0 | 0 | 0 | 228 |

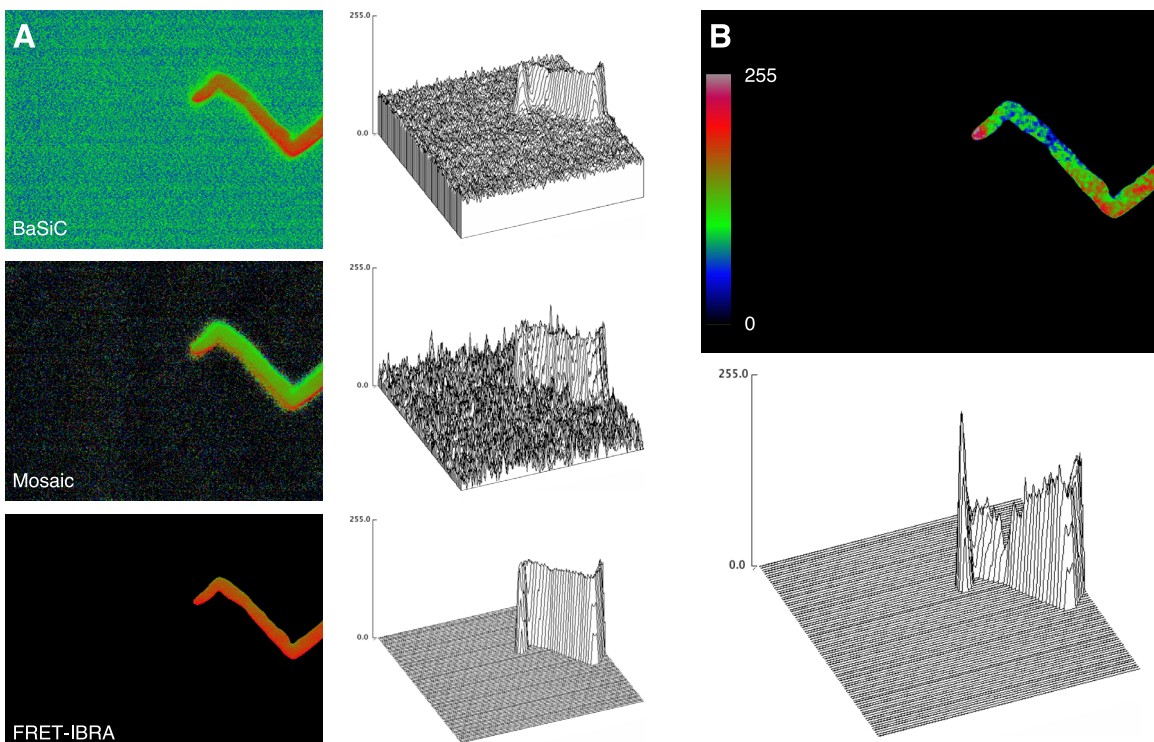

**Fig 4. Ratio image processing.** A) Background-subtracted acceptor and donor images from BaSiC-, Mosaic-, and FRET-IBRA-produced ratio images were processed and compared. Surface plots created from the ratio images give a better overview of the background subtraction quality and the signal-to-noise ratio achieved with each software (top: BaSiC, middle: Mosaic, bottom: FRET-IBRA). B) FRET-IBRA automatically registers and rescales the ratio image with robust rescaling to reveal the entire range of the signal spectrum.

around the pollen tube outline. In contrast to BaSiC and Mosaic, ratio images produced after FRET-IBRA background subtraction displayed a low uniform background signal, leading to a superior signal-to-noise ratio without strong artifacts and extreme outliers (Fig 4A, bottom panel).

From the FRET-IBRA processed image in Fig 4A, it is evident that mapping the full 8-bit signal intensity range is not ideal for visually representing the dynamics of the displayed calcium levels. The reason is that even a single outlier with an unexpectedly high value would determine the scale factor of the entire image. The equivalent also occurs at the lower end of the foreground signal range, leading to a relatively narrow intensity histogram and a flat image. Therefore, FRET-IBRA uses robust rescaling (10th—90th percentile) on the ratio image intensity values to reveal the hidden dynamics of the signal intensity. The registered and rescaled version of the ratio image clearly showcases a high distribution of calcium ions at the tip with a sharp drop-off outside the tip region. As expected, the informative quality of a ratio image hinges largely on achieving a robust signal-to-noise ratio from the input stacks, which is, in turn, highly dependent on the background subtraction algorithm.

FRET-IBRA is a fully-parallelized, modular, and flexible tool that provides a complete workflow solution to conduct ratiometric analyses of FRET images. The package has been shown to correct shading, effectively subtract background pixel intensities, and smoothen background noise more effectively than comparable packages for images with few objects of interest. In addition, FRET-IBRA creates ratiometric images and performs photobleaching

correction in a modular fashion, allowing for efficient parameter tuning for large image stacks. It should be noted that bleaching correction must be used with caution as it can cause artifacts, such as masking of true signal intensity variations or overcorrection, resulting in a false increase in signal intensity if the detrending methods provided are not appropriate. Therefore, it may sometimes be advisable to accept overall attenuation of the resulting ratio signal in order to detect short-term fluctuations. Alternatively, bleaching correction can be performed using other detrending algorithms outside of FRET-IBRA. The resultant images produced by FRET-IBRA on FRET images of growing pollen tubes have been shown to have high signal-to-noise ratios and no significant artifacts, allowing for accurate further quantitative or qualitative downstream analyses. Furthermore, the flexible nature of FRET-IBRA enables discontinuous frames to be easily processed and corrected individually, resulting in potentially significant time savings. FRET-IBRA performs well with various fluorophores tested, including biosensor proteins and fluorescent dyes. Background subtraction can also be performed using single stacks.

## Future directions

While FRET-IBRA is primarily built to be a command-line tool using configuration file-based parameters, the classes implemented in Python3 are modular and can be easily integrated into a larger analysis pipeline. Further developments are currently under way to locate regions of high ratiometric intensity and track their time-varying dynamics.

## Author Contributions

**Conceptualization:** Gautam Munglani, Hannes Vogler.

**Data curation:** Hannes Vogler.

**Formal analysis:** Gautam Munglani.

**Funding acquisition:** Ueli Grossniklaus.

**Investigation:** Hannes Vogler.

**Software:** Gautam Munglani.

**Supervision:** Ueli Grossniklaus.

**Visualization:** Hannes Vogler.

**Writing – original draft:** Gautam Munglani, Hannes Vogler.

**Writing – review & editing:** Gautam Munglani, Hannes Vogler, Ueli Grossniklaus.

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
