## [Decision Letter · Decision Letter 0]

25 Aug 2021

Dear Dr. Vogler,

Thank you very much for submitting your manuscript "Fast and flexible processing of large FRET image stacks using the FRET-IBRA toolkit" for consideration at PLOS Computational Biology.

As with all papers reviewed by the journal, your manuscript was reviewed by members of the editorial board and by several independent reviewers. In light of the reviews (below this email), we would like to invite the resubmission of a significantly-revised version that takes into account the reviewers' comments.

We cannot make any decision about publication until we have seen the revised manuscript and your response to the reviewers' comments. Your revised manuscript is also likely to be sent to reviewers for further evaluation.

Sincerely,

Dina Schneidman-Duhovny

Software Editor

PLOS Computational Biology

Dina Schneidman-Duhovny

Software Editor

PLOS Computational Biology

Reviewer's Responses to Questions

**Comments to the Authors:**

Reviewer #1: The data analysis pipeline presented in “Fast and flexible processing of large FRET image stacks using the FRET-IBRA toolkit” tackles major issues in obtaining robust quantitative estimates from fluorescence microscopy time series, especially when dealing with ratiometric reporters. The authors employ solid image processing techniques that can: a) subtract the background for each frame even with spatial variation; b) perform photobleaching correction; c) output a scaled ratio image that could be used for further quantitative analysis. This open-source tool has a clear and flexible methodology built on freely available and well-written Python code, which should appeal directly to the research of single cell growth/migration.

FRET-IBRA has the potential to become the go-to tool for researchers in the pollen tube and root hair community analyzing ratiometric data, which I believe would be enhanced if the authors address issues on three fronts: 1) benchmark; 2) automation; 3) distribution/interface. Although FRET-IBRA seems to offer a superior output than other related tools, this comparison was made with a single illustrative example and could benefit from directly quantifying variables of biological interest across multiple data-sets. Furthermore, automating the choice of certain parameters (mainly epsilon) could offer more objective criteria and minimize manual intervention. Lastly, despite providing a clear code and installation instructions in GitHub, the current distribution and interface requires a moderate level of programming experience that limits the potential users.

Regardless of the potential shortcomings, FRET-IBRA is certainly a useful tool that I personally intend to incorporate in my analyses. Below I further discuss points that can, hopefully, increase the adoption of this pipeline.

MAJOR POINTS

1) Benchmark

The authors use an example dataset to illustrate the workflow of FRET-IBRA and to compare it with other tools. While the representative dataset did provide a clear explanation of the method and its performance, it is difficult to evaluate how generalizable are the results when dealing with different datasets. Does the difference in performance with other tools hold up with different fluorescent reporters, microscope settings, image and object sizes? Relying on manual choice of parameters further complicates this matter (see discussion on automation), which is not to say that a manual option is not welcome. Furthermore, since the aim is to provide robust quantitative imaging, ideally one should also compare estimates of quantities-of-interest such as tip fluorescence ratio and tip-to-shank gradient. Showing the potential of FRET-IBRA to provide new biological insights from reliable spatiotemporal quantification of ratiometric fluorescence would greatly increase its impact within the aims of PLOS Computational Biology.

2) Automation

Although the possibility of correcting individual frames may be interesting when discontinuous events occur, requiring visual inspection and manual parameter choices compromises objectivity and increases the time required to analyze each series. How could one ensure comparable results when drastically different parameters are used to process different frames, channels and series?

I encourage the authors to provide an automatic choice of epsilon (and maybe even window size), which even if just “guesstimates” may help users to tune them as little as possible. Perhaps a few simplifying assumptions can aid in that endeavor, for example assuming that the number and location of foreground and background tiles should be very similar (if not identical) between channels.

Finally, while I fully appreciate that some experiments may lead to outliers, overcorrecting for them may lead to greater artifacts than simply disregarding specific time points. On that note, I would like to understand more the reasoning behind using image registration between acceptor and donor channels. Unless both channels are sampled with a large time interval, it seems to me that they should be nearly or completely identical. Thus, allowing for image registration – even if only a rigid linear transformation – seems to increase the chance of introducing artifacts. Although the online tutorial does mention that registration should be used only after choosing an “optimal” value of epsilon, there is seemingly no objective criteria for that. I can picture scenarios where an experimental intervention may displace the cell, where registration would be important to match its position, however this would not be between channels but between consecutive frames. I ask the authors to consider adding further explanation or a discussion on the matter.

3) Distribution/interface

The choice of implementing the FRET-IBRA pipeline in Python has multiple benefits but also imposes challenges to users without a moderate programming experience. Even for initiates, some of the requirements of the pipeline are notoriously cumbersome to install. Despite the clear GitHub instructions, in my personal experience the installation took considerable time and effort. Even after managing the installation, there were errors in the ratiometric module and in processing 1024x1024 images that I am unsure whether they stem from issues in the code or simply installation flaws (see attached report). Two of the datasets tested yielded apparently great results in terms of background subtraction in both channels, however the ratiometric module yielded the error: “IndexError: cannot do a non-empty take from an empty axes” (see report). The other two datasets with 1024x1024 images were not processed due to a “ValueError: cannot reshape array of size 1048576 into shape (40,25,newaxis,25)” (see report).

Regardless of my personal experience testing the pipeline, a few solutions would definitely attract more users. The most inclusive option would be a web-interface where the user could interactively tune parameters. A Python-based graphical user interface could also help attract users, but it would likely still require cumbersome installations. An interactive Jupyter notebook can be a great choice but still may require a little programming skill. I encourage the authors to consider what is their audience and what would be the friendliest (and yet achievable) solution.

Finally, if sufficient automation is achieved, it would be interesting to have the complete workflow performed sequentially for a given series with a single command (background subtraction for each series, ratiometric calculation and optional photobleaching correction). Ideally a “batch mode” could ensure that a complete data cohort is analyzed with comparable parameters, although it may still be desirable to maintain a level of intervention in specific frames.

MINOR POINTS

- Lacking a discussion on how correcting photobleaching can insert artifacts.

- The standard deviation can be arbitrarily reduced with the amount of smoothing used (Pg. 5 ln 112-113). This maybe the reason why we see a decrease in the standard deviation and maximum values with window size in table 1. Thus, it maybe not be completely fair to use the standard deviation to compare with methods that do not perform smoothing.

- Pg. 6 ln 132 “rendering any quantitative analysis impossible” is a strong assertion, I recommend softening it.

- Pg. 6 ln 146-147 “true dynamics of the signal intensity” is a strong assertion, I recommend substituting “true” by “hidden” or other equivalent term.

- Pg. 7 ln 150 “best possible signal-to-noise ratio” is a strong assertion, I recommend softening it.

Reviewer #2: Summary:

The paper proposes a processing tool for FRET image stacks, including i) background subtraction for acceptor and donor time-lapse images, ii) registration between acceptor and donor movies to produce ratio images and iii) photobleaching correction. It can be used to quantify the spatial distribution of calcium ions during pollen tube growth under mechanical constraints and the code is released under BSD license.

Generally, it is not a methodologically innovative paper, as the primary methods involved in the tool are based on existing methods. For example, background subtraction is based on DBSCAN clustering proposed in another paper. Registration between acceptor and donor movies are a standard rigid-body registration. Nevertheless, a tool that comprises multiple processing steps could be still useful in practice is if its usability and robustness are sufficiently demonstrated. Hence, I would suggest authors should consider evaluating their tool more thoroughly, use multiple movies, best captured by different microscopy under different conditions, or in multiple labs, to demonstrate the generalisability of the proposed tool. These issues should be addressed in the revision.

Major issues:

i) The background subtraction module is based on a DBSCAN clustering algorithm on image tiles to separate foreground and background, proposed in Schwarzfisher et al (2011). Since this clustering needs to be done for each frame in the movie and is generally slow, does the proposed tool improve the original algorithm in terms of speed?

ii) Comparison between BaSiC: as far as I know, BaSiC has a function to correct background (by setting Fiji plugin “Temporal_drift of baseline” to be “replace with zero” and it has a zero-like background by then. So it is surprising that your BaSiC corrected movie still has a high background value. Nevertheless, BaSiC might have a problem with highly-correlated foreground, which seems to be the case of the present example (exemplary movie in the github example) and may not function properly (as that breaks the basic assumption of BaSiC)

iii) How many movies are in the evaluation set. It seems to me only one pair of movies, i.e. acceptor and donor, is used. In my opinion, that is not sufficient to evaluate the robustness of the tool, which is also my main criticism of the paper.

Suggestions

i) If authors can also provide some biological-relevant down-stream analysis, they would be a more solid demonstration of the necessity of such a tool.

ii) Authors should also discuss the potential usage and limitation of the tool: when it works and when it could fail, to prevent misuse of the tool.

**Have the authors made all data and (if applicable) computational code underlying the findings in their manuscript fully available?**

Reviewer #1: **No: **The code is fully available on GitHub with an example data set. However, the actual data set used to compare FRET-IBRA with other tools was not (probably due to size constrains).

Reviewer #2: Yes

PLOS authors have the option to publish the peer review history of their article (what does this mean?). If published, this will include your full peer review and any attached files.

Reviewer #1: **Yes: **Daniel Damineli

Reviewer #2: No
---

## [Decision Letter · Decision Letter 1]

2 Jan 2022

Dear Dr. Vogler,

Thank you very much for submitting your manuscript "Fast and flexible processing of large FRET image stacks using the FRET-IBRA toolkit" for consideration at PLOS Computational Biology. As with all papers reviewed by the journal, your manuscript was reviewed by members of the editorial board and by several independent reviewers. The reviewers appreciated the attention to an important topic. Based on the reviews, we are likely to accept this manuscript for publication, providing that you modify the manuscript according to the review recommendations.

Sincerely,

Dina Schneidman

Software Editor

PLOS Computational Biology

[LINK]

Reviewer's Responses to Questions

**Comments to the Authors:**

Reviewer #1: OVERALL REMARKS

Munglani et al. addressed several points raised by the reviewers and presented: 1) a friendlier interface to the FRET-IBRA pipeline both in terms of user input (GUI) and workflow (complete background subtraction on both channels and ratiometric processing); 2) more examples of its biological applicability in the GitHub repository and 3) improved presentation/discussion of the results. These changes enhance the usability of the tool, although there are still a few shortcomings in the presentation and discussion of the results pointed out by both reviewers in the previous considerations. In my opinion, the major limitation of the manuscript still is to provide a clear demonstration of the biological applicability of the pipeline, which should be easily achievable by including long ratiometric movies and time series. I suggest illustrating the potential of FRET-IBRA in reporting the known oscillations in cytosolic calcium concentrations at the pollen tube tip. I reckon such addition to the manuscript should be easily achievable with data from the authors themselves, otherwise I would be happy to provide a suitable data set.

DETAILED REMARKS

I am glad the authors took the effort to extend FRET-IBRA’s interface by providing a GUI and the option to do background subtraction for both channels and calculate the ratio, these changes should facilitate the accessibility of the pipeline for many users. Furthermore, the biological examples in the GitHub repository suggest a wide range of applicability of the tool although the data was not explicitly incorporated in the manuscript. Thus:

- Discussions about the examples in GitHub are limited since details are lacking

- Examples are single time points making it hard to evaluate the potential of the pipeline to generate a long ratiometric movie

- The example in Brachypodium roots suggest strong spatial artifacts introduced by the background subtraction, especially of the acceptor channel (see figure in the attached report)

- It is often difficult to interpret the biological relevance of the ratiometric output (e.g. Brachypodium roots example) without a clear reference, whereas studying its temporal behavior can provide greater reliability to the result

Regardless of other examples, the model system already presented in the manuscript (i.e. pollen tubes) is ideal to show whether the ratiometric processing can produce biologically meaningful results. Pollen tube growth is accompanied by notorious tip-focused oscillations in cytosolic calcium, which could be used as a proof-of-principle that the method can adequately capture biological oscillations.

MINOR POINTS

- I am still unable to produce ratiometric stacks using 16 bit images, even when using the suggested nwindow (or others I tried). Admittedly this can be some issue on my end or something extremely simple to fix (check log in the attached report)

- I suggest adding an option to use bg/fg tiles from one channel to the other when image registration is not being used. This makes sense given the pixelwise stoichiometry assumption and would cut processing time nearly in half while avoiding artifacts due to different bg/fg definition in acceptor and donor channels. I emphasize that this is not a mandatory suggestion.

- One idea is to include loess as a method for photobleaching correction, depending on the series length a sufficiently large span parameter should capture the slow trend. One idea is to use degree 1 only to avoid inserting artifacts in the trend. I emphasize that this is not a mandatory suggestion.

- Sentence in the abstract stating that the package has been “extensively used in quantifying the spatial distribution of calcium ions during pollen tube growth under mechanical constraints.” has not been discussed in the text and no direct data was shown. As it is formulated it sounds that FRET-IBRA has been extensively used in the community rather than on a particular data set. Please reformulate (e.g. specify the lab) or remove.

- Ln 11. “cleanest signal possible” I recommend softening the assertion.

- Ln 145. “without any artifacts” I suggest replacing by something like “without clear artifacts or extreme outliers”

**Have the authors made all data and (if applicable) computational code underlying the findings in their manuscript fully available?**

Reviewer #1: Yes

PLOS authors have the option to publish the peer review history of their article (what does this mean?). If published, this will include your full peer review and any attached files.

Reviewer #1: **Yes: **Daniel Damineli

Figure Files:

Data Requirements:

Reproducibility:

References:

---

## [Decision Letter · Decision Letter 2]

16 Feb 2022

Dear Dr. Vogler,

We are pleased to inform you that your manuscript 'Fast and flexible processing of large FRET image stacks using the FRET-IBRA toolkit' has been provisionally accepted for publication in PLOS Computational Biology.

Before your manuscript can be formally accepted you will need to complete some formatting changes, which you will receive in a follow up email. A member of our team will be in touch with a set of requests. Also please address the comments of Reviewer 1 (below) in the final version.

Best regards,

Dina Schneidman

Software Editor

PLOS Computational Biology

Reviewer's Responses to Questions

**Comments to the Authors:**

Reviewer #1: In this revised version of the manuscript, the authors addressed the main shortcomings raised in the previous revision by applying the pipeline to pollen tube oscillations data and by fixing minor code issues. The present version of the paper and GitHub code offers the cell biology community a significant tool for processing ratiometric data. However, there are still minor points I would like to raise, although I believe they should be implemented at the author’s discretion.

REMAINING REMARKS

Given the significant improvement and number of revised versions the authors have already provided, I emphasize the points below are suggestions that could be implemented if the authors find it in their best interest.

1) Oscillations:

I was hoping the authors would use pollen tube tip oscillations as means to demonstrate the performance of FRET-IBRA to reveal biologically relevant phenomena. As such, ideally, I was expecting to see an additional figure in the paper showing the oscillation extracted with FRET-IBRA preferably in comparison to oscillations extracted with other algorithms. Instead, the authors chose to include part of the result in the GitHub repository, which not only limits the size of the output that can be shared but also does not highlight the performance of FRET-IBRA in terms of revealing clear oscillations. I encourage the authors to show “tip fluorescence vs time” plots instead of the raw output of the pipeline.

2) Limitations of the background subtraction algorithm:

In the previous revision I raised a question about spatial artifacts introduced by the background subtraction algorithm in Brachypodium roots data (specifically “Root_acceptor_back.tif” and “Root_donor_back.tif” found in the GitHub repository). It is clear there are spatial artifacts and this should be discussed, as users have to be aware of this possibility. As such, I did not understand the author’s answer that “Background subtraction, however, works still well, even with very low signals”. I recommend these specific files are inspected carefully and either: a) adequate parameters are used as to not produce spatial artifact; b) they are used as a cautionary example for users (which should be made clear in the text); c) they are excluded altogether.

3) Additional information on the GitHub repository:

I appreciate the authors included README files in GitHub examples, although they should include more explicit information about organism used and source of the data.

**Have the authors made all data and (if applicable) computational code underlying the findings in their manuscript fully available?**

Reviewer #1: Yes

PLOS authors have the option to publish the peer review history of their article (what does this mean?). If published, this will include your full peer review and any attached files.

Reviewer #1: **Yes: **Daniel Damineli

---

## [Editor Report · Acceptance letter]

31 Mar 2022

PCOMPBIOL-D-21-01180R2 

Fast and flexible processing of large FRET image stacks using the FRET-IBRA toolkit

Dear Dr Vogler,

I am pleased to inform you that your manuscript has been formally accepted for publication in PLOS Computational Biology. Your manuscript is now with our production department and you will be notified of the publication date in due course.

With kind regards,

Agnes Pap
